# Treating Cardiovascular Disease in the Inflammatory Setting of Rheumatoid Arthritis: An Ongoing Challenge

**DOI:** 10.3390/biomedicines12071608

**Published:** 2024-07-19

**Authors:** Saloni Godbole, Jenny Lue Solomon, Maryann Johnson, Ankita Srivastava, Steven E. Carsons, Elise Belilos, Joshua De Leon, Allison B. Reiss

**Affiliations:** Department of Medicine and Biomedical Research Institute, NYU Grossman Long Island School of Medicine, Mineola, NY 11501, USA; sgodbole22@gmail.com (S.G.); jenny.lue@nyulangone.org (J.L.S.); johnson.maryann10@gmail.com (M.J.); ankita.srivastava@nyulangone.org (A.S.); steven.carsons@nyulangone.org (S.E.C.); elise.belilos@nyulangone.org (E.B.); joshua.deleon@nyulangone.org (J.D.L.)

**Keywords:** cardiovascular disease, rheumatoid arthritis, therapeutics, statin, inflammation, biological therapy

## Abstract

Despite progress in treating rheumatoid arthritis, this autoimmune disorder confers an increased risk of developing cardiovascular disease (CVD). Widely used screening protocols and current clinical guidelines are inadequate for the early detection of CVD in persons with rheumatoid arthritis. Traditional CVD risk factors alone cannot be applied because they underestimate CVD risk in rheumatoid arthritis, missing the window of opportunity for prompt intervention to decrease morbidity and mortality. The lipid profile is insufficient to assess CVD risk. This review delves into the connection between systemic inflammation in rheumatoid arthritis and the premature onset of CVD. The shared inflammatory and immunologic pathways between the two diseases that result in subclinical atherosclerosis and disrupted cholesterol homeostasis are examined. The treatment armamentarium for rheumatoid arthritis is summarized, with a particular focus on each medication’s cardiovascular effect, as well as the mechanism of action, risk–benefit profile, safety, and cost. A clinical approach to CVD screening and treatment for rheumatoid arthritis patients is proposed based on the available evidence. The mortality gap between rheumatoid arthritis and non-rheumatoid arthritis populations due to premature CVD represents an urgent research need in the fields of cardiology and rheumatology. Future research areas, including risk assessment tools and novel immunotherapeutic targets, are highlighted.

## 1. Introduction

Rheumatoid arthritis (RA) is a chronic autoimmune disorder characterized by progressive inflammation in the synovial joints and extra-articular structures [1,2]. It is a systemic disease that can severely impair mobility and quality of life. Approximately 1% of the global population is affected [3,4,5,6]. The peak incidence of RA is between the ages of 30 and 50 years, and females are 2 to 3 times more commonly affected than males [7,8].

The incidence of coronary artery atherosclerosis is higher in persons with RA, making cardiovascular disease (CVD) a significant cause of morbidity and mortality for this patient group [9,10,11]. Individuals with RA have a 50% increased risk of developing CVD compared to the general population, and it has been estimated that persons with RA above age 45 have an almost three-fold higher risk of CVD death [12]. The prevalence of subclinical atherosclerosis is higher in RA patients than in healthy persons, and this difference remains after adjusting for traditional CVD risk factors [13]. In addition to sub-clinical atherosclerosis, CVD in RA may also manifest as silent ischemia and silent myocardial infarction [14,15]. Unfortunately, the accurate prediction of CVD in RA is difficult, and the measurement of Framingham risk factors or RA disease activity markers underestimate this risk [16,17,18,19]. The lack of biomarkers has impeded efforts to detect CVD before it manifests when preventive therapy or lifestyle modification could be instituted [20,21]. Further, clinical guidelines offer little direction in the selection of tailored therapeutics for the prevention and treatment of CVD in the setting of RA.

The purpose of this narrative review is to report the present state of understanding of how to best manage CVD and minimize CVD risk in persons diagnosed with RA. This review covers the effects of commonly used anti-rheumatic, atheroprotective, and lipid-lowering drugs on heart health in RA patients. A pragmatic approach to clinical care of this atherosclerosis-prone population is discussed.

## 2. Pathophysiology of Cardiovascular Diseases

CVD is an umbrella term for a group of diseases affecting the heart and blood vessels. There are many different types of CVD, which include coronary artery disease, heart failure, valvular heart disease, arrhythmia, peripheral artery disease, aortic disease, and deep vein thrombosis [22,23]. CVD is the leading cause of death globally. The incidence of CVD increases with increases in plasma cholesterol levels, arterial stiffness, and peripheral vascular resistance [24,25].

Atherosclerosis is the major cause of CVD-related death worldwide. Atherosclerosis is a chronic disease characterized by increased deposition of lipids, fibrous elements, and calcium within large arterial walls [26]. This process is triggered by damage to the endothelial cell monolayer of the arterial wall, resulting in endothelial activation. This further stimulates changes in transendothelial permeability and accumulation of lipids, particularly particles of low-density lipoprotein (LDL) cholesterol into the intima of the arterial wall [27]. Modified forms of LDL such as oxidized LDL also amass in the vascular wall and are strong chemoattractants for monocytes [28,29]. The activated endothelium expresses adhesion molecules, which also recruit monocytes that can then migrate into the subendothelial space, where they differentiate into resident macrophages and avidly take up modified LDLs, transforming into foam cells [30,31].

Oxidized LDL enters macrophages primarily via CD36 scavenger receptors. Internalized oxidized LDL can then induce pro-inflammatory cytokine and chemokine expression [32]. These processes lead to macrophage retention of a massive burden of intracellular cholesterol leading to macrophage foam cell transformation [33]. The foam cells produced will then become a crucial constituent of the fatty streak that forms in the subendothelial layer of the vessel wall and represents early atherosclerosis [34,35]. The activated macrophage secretion of various inflammatory cytokines and growth factors draws more circulating monocytes to the atherosclerotic lesion and stimulates smooth muscle cell proliferation and migration [36]. These processes constitute a summary of the major steps in atherosclerotic plaque pathogenesis.

## 3. Mechanisms and Role of Inflammation in Cardiovascular Disease (CVD)

The pathophysiology of RA involves inflammatory cascades, which lead to persistent synovial inflammation and eventually damage to surrounding cartilage and bone [37]. The same inflammatory cascades that lead to joint inflammation and destruction also contribute to increased atherosclerotic burden in patients with RA. Measures of disease activity in RA do not correlate directly with cardiovascular risk, and RA disease measures are not reliable for the prediction of CV events or mortality [38,39,40]. However, there are some markers with potential utility. For instance, the overexpression of tumor necrosis factor (TNF)-like cytokine 1A in the synovium of rheumatoid arthritis patients correlates with both disease activity and atheromatous plaque height and the formation of new plaques [41].

Cytokines are a key player in the inflammatory process and contribute to cardiovascular risk by recruiting immune cells, activating the endothelium, differentiating monocytes, forming foam cells, and causing plaque rupture and thrombosis [42,43]. In fact, the pro-inflammatory cytokines involved in the pathogenesis of atherosclerosis and RA have substantial overlap [44,45]. TNF-α, IL-6, and IL-1β are examples of cytokines involved in both RA and atherosclerotic CVD [46]. Interactions between T and B lymphocytes, fibroblasts, and macrophages lead to the overproduction of TNF-α, which in turn stimulates other cytokines known to promote inflammation, including IL-1β and IL-6 [47]. TNF-α drives endothelial dysfunction, one of the earliest stages of plaque formation in atherosclerosis [48]. It also lowers nitric oxide and thrombomodulin levels, creating a prothrombotic environment [49,50,51]. IL-1β provokes endothelial cells to upregulate surface adhesion molecule expression, thereby bringing about macrophage recruitment to the arterial intima [52]. Once the endothelial layer has been breached by inflammatory cells, cholesterol buildup occurs inside the artery. Monocytes are recruited into the subendothelial space, where they differentiate into macrophages, internalize oxidized lipoproteins, and transform into atherogenic foam cells [53]. As foam cells gather in particular locations in the arterial intima, they form a fatty streak, a precursor to the atherosclerotic plaque. Another key cytokine involved in all stages of atherogenesis is IFN-y. In addition to damaging vasculature via oxidative stress, IFN-y leads to plaque destabilization via smooth muscle cell apoptosis and the release of matrix metalloproteinases [54,55]. Not all cytokines involved in atherosclerosis are pro-inflammatory, however. TGF-β and IL-10 exert an atheroprotective role when released by regulatory T-cells [56,57]. IL-10 has been shown to have anti-inflammatory properties, which decelerate the progression of atherosclerosis by inhibiting inflammation and cell apoptosis [58]. Murine models of acute myocardial infarction treated with IL-10 demonstrated decreased inflammation and improved left ventricular function [59]. Conversely, atherosclerosis was accelerated in vivo with genetic inactivation of IL-10 [45]. Orecchioni et al. used the apolipoprotein (apo)E-deficient mouse model of atherosclerosis to show that resident and inflammatory macrophages in the aorta are major producers of IL-10 [60]. They then generated apoE-deficient mice with myeloid-specific inactivation of IL-10 and demonstrated that these mice exhibit a pro-inflammatory state, with significantly increased expression of TNF-ɑ, and CCL2, a chemokine linked to atherosclerosis as well as a trend toward increased IL-1β. This resulted in significantly increased total cholesterol levels and, when these mice were fed a Western diet, a five-fold increase in the size of atherosclerotic lesions.

The inflammatory pathways involved in RA not only cause endothelial activation as a route to cholesterol accumulation within vasculature but also disrupt cholesterol homeostasis, further increasing cardiovascular risk [61,62] (Figure 1). Cholesterol equilibrium in macrophages is maintained by three processes: cholesterol ingress through designated receptors, intracellular degradation by enzymes, and cholesterol movement out of the cell. Cell culture experiments in which human macrophages are incubated in human plasma have found that the plasma of persons diagnosed with RA causes a diminution in the level of macrophage cholesterol efflux proteins while also upregulating scavenger receptors that influx cholesterol. This pattern amplifies the macrophage cholesterol burden and foam cell formation [63]. The pervasive inflammatory milieu that exists in the RA patient influences the level and function of both low-density lipoprotein (LDL) and high-density lipoprotein (HDL). Studies have shown that when exposed to inflammation and oxidative stress, HDL can become pro-atherogenic rather than atheroprotective [64,65,66]. During acute or chronic inflammation, the HDL structure changes to include oxidized lipids and acute phase reactants such as serum amyloid A and ceruloplasmin, rather than cholesterol-transport proteins such as apoA-1 and antioxidant enzymes [67]. This pro-inflammatory HDL, or piHDL, results in impaired reverse cholesterol transport, decreased lipoprotein synthesis, and increased oxidation of LDL. RA patients were found to have a higher percentage of the most electronegative subfraction of HDL, which correlated with subclinical atherosclerosis [68]. Further confirming the relationship of RA to abnormal HDL, HDL from RA patients with high disease activity was shown to have a decreased ability to promote cholesterol efflux compared to HDL from patients with low disease activity or in clinical remission [69]. These researchers also found a significant inverse correlation of the erythrocyte sedimentation rate (ESR) with cholesterol efflux after controlling for confounding variables including diabetes, glucocorticoid use, and smoking. Paraoxonase (PON)-1, a glycoprotein located in the HDL particle, is responsible for the antioxidant properties of HDL. In RA patients, both the concentration of PON-1 and its enzymatic activity are reduced, and this reduction correlates with an increased carotid plaque burden found using ultrasound [70].

Inflammation-related disturbance in HDL functions, such as the inhibition of LDL oxidation and reverse cholesterol transport, may help explain why traditional lipid profile evaluations are insufficient for predicting cardiovascular risk for RA patients. Not only is the composition of HDL affected in RA, but LDL cholesterol is also different, with more citrullinated and homocitrullinated forms detected in RA patients than in healthy subjects [71]. Citrullinated and homocitrullinated LDL has greater atherogenic potential and has been shown to induce foam cell formation in cultured macrophages [72].

## 4. Current Treatment Options

The therapeutic armamentarium in RA has varying pro- and anti-atherogenic properties. Some agents, such as corticosteroids and non-steroidal anti-inflammatory drugs (NSAIDs), are well known to enhance atherosclerotic risk [73]. For others, such as methotrexate and biologics, the evidence is accumulating that certain agents may be beneficial for both RA management and risk attenuation for atherosclerosis. The following subsections describe recent scientific data on the most commonly used agents for RA treatment (summarized in Table 1).

### 4.1. Methotrexate

Methotrexate is the cornerstone of RA treatment [74]. The mechanism of action of methotrexate is the inhibition of the enzyme dihydrofolate reductase, which prevents the purine and pyrimidine formation needed for DNA and RNA synthesis [75]. Methotrexate offers protection against CVD both indirectly, through the reduction in overall inflammation, and directly via anti-atherogenic effects. Methotrexate has been shown to promote the outflow of cholesterol from macrophages, the enhancement of free radical scavenging, and the improvement in endothelial function [47]. Levels of adenosine, an endogenous nucleoside with many cardioprotective qualities, are increased by methotrexate [76,77]. Cardioprotective effects of adenosine include coronary artery vasodilation, the reduction in myocardial oxygen demand, and anti-platelet activity. Through the increased generation of adenosine with consequent adenosine A2A receptor activation, methotrexate promotes reverse cholesterol transport and limits foam cell formation in cultured THP-1 human macrophages, a widely used cellular model for macrophage behavior during atherosclerosis [78,79]. Furthermore, methotrexate treatment is associated with reduced risk of type 2 diabetes by improving insulin sensitivity via the increased extracellular adenosine concentration, promoting metabolism and transport of glucose [80]. As it is well-established that patients with diabetes are at higher mortality risk from CVD, treatment with methotrexate may have further value. The numerous cardioprotective mechanisms wielded by methotrexate add up to a real-world, tangible benefit for patients with RA. By some estimates, methotrexate use is associated with a 70% reduction in cardiovascular mortality and a 60% reduction in all-cause mortality [81]. Methotrexate treatment was shown to reduce cardiovascular risk in RA in a prospective cohort study using the Veterans Affairs Rheumatoid Arthritis (VARA) registry [82]. Data obtained on 2044 US veterans with RA showed that the lower CVD risk associated with methotrexate was independent of RA disease activity, age, body mass index, and traditional CVD risk factors such as hypertension, hyperlipidemia, and diabetes. These beneficial cardiovascular effects, along with their lower cost, established the effectiveness and safety profile that underscore methotrexate’s role as an initial treatment for RA patients [68].

### 4.2. Non-Steroidal Anti-Inflammatory Drugs (NSAIDs)

NSAIDS are frequently employed as an initial or adjunctive strategy in a regimen designed to relieve the symptoms of pain and stiffness in RA and other rheumatologic conditions. The mechanism of action of NSAIDs is via the inhibition of vasodilatory cyclooxygenase (COX), which reduces pain and inflammation through the inhibition of prostaglandins [83,84]. COX-2 inhibitors preserve platelet function, thus favoring thrombosis. Even short-term use of NSAIDs has been demonstrated to raise the risk of cardiovascular events and stroke. A meta-analysis of 34 papers, with 28 focused on patients with RA and 6 directed at patients with psoriasis or psoriatic arthritis, found that CVD risk was higher for both groups with either non-selective or COX-2 NSAIDs [85]. Though the exact mechanism has not yet been determined, cell culture studies in THP-1 human macrophages show that the COX-2-selective inhibitor NS398 enhances lipid-overload and foam cell transformation via the downregulation of the reverse cholesterol transport proteins cholesterol 27-hydroxylase and ATP-binding cassette transporter (ABC)A1 [86]. In cultured human macrophages, celecoxib and rofecoxib increased foam cell formation, likely via the suppression of the cholesterol efflux proteins 27-hydroxylase and ABCA1 [87]. Celecoxib also upregulated CD36, a scavenger receptor responsible for cholesterol uptake, which would add to the macrophage lipid burden. Therefore, NSAIDs such as COX-2 inhibitors like celecoxib can both benefit RA patients while also negatively impacting lipid metabolism via interference with the transport of cholesterol. The Standard Care versus celecoxib Outcome Trial (SCOT) randomized 7297 persons over age 60 years with either RA or osteoarthritis to continue on their previously prescribed non-selective NSAID or switch to celecoxib, and after a median follow-up of 3 years, outcomes were comparable, with similar cardiovascular event rates in both COX-2 and non-selective NSAID treatment groups [88]. Though NSAIDs help manage RA symptoms, they are also known to cause gastrointestinal events, including iron deficiency anemia, and they carry the risk of nephrotoxicity, which is a consideration for certain patient populations. In the Prospective Randomized Evaluation of Celecoxib Integrated Safety vs. Ibuprofen or Naproxen (PRECISION) trial, the group of patients taking celecoxib had significantly fewer gastrointestinal events than the naproxen or ibuprofen groups, thus other comorbidities should be considered when choosing an NSAID for treating RA [83].

### 4.3. Leflunomide

The mechanism of action of leflunomide is the inhibition of dihydroorotate dehydrogenase, an enzyme essential for pyrimidine synthesis and, therefore, the proliferation of activated T-lymphocytes [89]. Direct associations between leflunomide and cardiovascular risk factors are not well-described. Leflunomide may indirectly affect cardiovascular risk via its effects on blood pressure, body weight, and plasma glucose concentrations. A post hoc analysis of 169 patients with RA showed that leflunomide use was associated with both decreased body weight and lower plasma glucose concentrations [90]. Ribosomal protein S6 kinase 1 (S6K1), which is a serine/threonine kinase downstream of mTOR, under high nutrition circumstances will remain activated, which induces insulin resistance and obesity; however, in mouse models, leflunomide inhibited S6K1 activity and promoted autophagy of adipocytes [91]. Leflunomide is associated with an increased incidence of hypertension. Drug-related hypertension occurs in approximately 2 to 4% of leflunomide-treated patients compared to 2% for comparator drugs including sulfasalazine and methotrexate [92]. Regarding safety profile, in a cohort study of RA patients taking leflunomide, severe infection occurred with an incidence of 3.3%, though these patients were also taking additional medications such as glucocorticoids or methotrexate, therefore making it difficult to assess leflunomide’s individual risk for infection [93]. While the most common side effect of leflunomide is gastrointestinal disturbance, hepatotoxicity is a concern, which is why it carries a black-boxed warning to monitor liver enzymes [89].

### 4.4. Sulfasalazine

The exact mechanism of action of sulfasalazine is not fully understood. Sulfasalazine is made up of sulfapyridine, an antibiotic, and 5-aminosalicyclic acid, an NSAID. It can inhibit platelet thromboxane synthetase and prevent arachidonic acid-mediated platelet aggregation, similar to the mechanism of aspirin [94]. Furthermore, it increases HDL, which is cardioprotective [95]. It also decreases inflammatory cytokines by inhibiting nuclear factor-κB (NF-κB), which promotes both inflammation and the adhesion that leads to monocyte recruitment in nascent atherogenic plaques. The inhibitory effect of sulfasalazine on NF-κB occurs due to the halting of the phosphorylation and breakdown of NF-κB’s inhibitory subunit IκB [47]. Of note, though the downregulation of TNF-α gene expression was observed in isolated mononuclear cells after exposure to sulfasalazine, this did not appear to correlate to changes in systemic inflammatory markers [96]. Sulfasalazine has also been associated with a small but significant reduction in cardiovascular events [97]. Sulfasalazine can have positive effects on the vasculature by inducing the enzyme HO-1, which, in turn, lowers reactive oxygen radicals and oxidative stress and may improve endothelial function in vessel walls [98,99,100]. Sulfasalazine is conditionally recommended by the American College of Rheumatology for patients with low disease activity over methotrexate because of its decreased immunosuppressive activity and for patients who wish to avoid the adverse effects of methotrexate [74].

### 4.5. Hydroxychloroquine

Hydroxychloroquine accumulates in and alkalinizes lysosomes, thereby inhibiting their functions [101]. Though the exact mechanism of action of hydroxychloroquine is unknown, it is thought to immunosuppress by blocking the proliferation of T-cells, halting macrophage cytokine production, and preventing toll-like receptor stimulation [102,103]. Hydroxychloroquine has been well-established to improve joint pain and prevent cartilage destruction, and it is safe to prescribe in pregnancy, though retinopathy is a concern for long-term use, for which patients should undergo regular ophthalmological screenings [104]. In a retrospective study of a cohort of 1266 RA patients, the risk of incident CVD was reduced by 72% with hydroxychloroquine treatment [105]. Part of its effect on preventing CVD may be in reducing cytokines like IL-1, IL-6, and TNF-α, which contribute to the formation of plaques [106]. Studies suggest that hydroxychloroquine is potentially anti-thrombotic via mechanisms involving platelet adhesion and interaction with coagulation factors [107]. In murine studies of RA by Shi et al., hydroxychloroquine both reduced atherosclerosis and increased microbiota flora in the gut [108]. The authors hypothesized that the atheroprotection was conferred due to decreased platelet aggregation and the cleaving of arachidonic acid from platelets as well as the suppression of antiphospholipid antibodies. The beneficial effects of hydroxychloroquine on the gut microbiota were attributed to an increase in *Akkermansia muciniphila*, which produces enzymes that decrease inflammation, produce an anti-atherosclerotic effect, and may prevent obesity. Additionally, hydroxychloroquine may improve outcomes in diabetic patients, mitigating a major risk factor for CVD. Hydroxychloroquine has been shown to increase the binding of insulin to its receptor, and it has also been shown to have a significant impact on increased insulin sensitivity [109].

### 4.6. Glucocorticoids

Glucocorticoids such as prednisone are a common treatment to suppress inflammation caused by RA. Glucocorticoids exert most of their effects by binding to intracellular glucocorticoid receptors, translocating to the nucleus, and changing gene expression [110]. They function as agonists of glucocorticoid and mineralocorticoid receptors, impacting ligand-activated transcription factors and directly affecting endothelial function [111]. However, their use may also increase the risk of cardiovascular events. They can promote oxidative stress and decrease nitric oxide (NO) in the body via downregulating endothelial NO gene transcription and expression as well as enhancing the production of reactive oxygen species, thereby diminishing NO bioavailability in the vasculature [111]. Yet, they are so frequently used due to their quick onset to relieve joint inflammation and pain as well as their affordable cost. Steroids have also been shown to increase cardiovascular events such as myocardial infarction, stroke, and heart failure in patients being treated for RA [85,112]. A study of steroid dose thresholds associated with all-cause and cardiovascular mortality revealed an incidence ratio of 1.89 for cardiovascular-related deaths associated with glucocorticoid use [113]. For these reasons, glucocorticoids are a clear example of the “double-edged sword” due to their swift therapeutic effect; therefore, from the available evidence, patients should be treated with the lowest dose for the shortest time possible to control the disease [114]. Moreover, glucocorticoids are well known to increase the risk of insulin resistance and hypertension [111,115,116]. For these reasons, glucocorticoids are an example of the “double-edged sword”, signifying that they have utility due to their swift therapeutic effect but also confer greater risk for CVD; therefore, patients should be treated with the lowest dose for the shortest time possible to control the disease [114]. Interestingly, in active RA patients, data are conflicting as to whether endothelial function is truly negatively impacted by glucocorticoids [111,115]. Confounding factors in prior studies, including the co-administration of other DMARDs, participants with a wide range of disease stages, and non-standardized dosages of glucocorticoids further affect the external validity of prior findings [111]. More research is necessary to clarify the effect of glucocorticoids on endothelial function, specifically in the setting of RA.

### 4.7. Janus Kinase Inhibitors

Janus kinase (JAK) inhibitors are non-receptor tyrosine kinases. Their mechanism of action is the transduction of cytokine signaling through the JAK signal transducer and activation of the Janus kinase-signal transducer and activator of transcription (JAK-STAT) transcription pathway, which in turn controls the transcription of genes that modulate inflammatory conditions [117].

The ORAL Surveillance randomized controlled trial comparing tofacitinib 5 mg twice daily, tofacitinib 10 mg twice daily, and the TNF inhibitor adalimumab for RA demonstrated an increased risk of major adverse cardiovascular events (MACE) with a hazard ratio of 1.33 [118]. This led the FDA to issue black-box warnings for tofacitinib, baricitinib, and upadacitinib. Sakai et al. also found increased CVD event risk in RA patients given JAK inhibitor therapy in a retrospective longitudinal study from Japan [119]. Real-world registries have generated conflicting results. A nationwide population-based cohort study of the French national health system did not show a significantly increased risk of MACE in patients exposed to JAK inhibitors [120].

JAK inhibitor treatment can lead to increased total cholesterol, LDL cholesterol, HDL cholesterol, and triglycerides, though it has not been shown to result in a significantly higher risk of cardiovascular events [114,121]. Tofacitinib has been shown to increase the levels of HDL and LDL more than TNF inhibitors, possibly by reversing cholesterol ester catabolism [122,123]. Low doses of atorvastatin reverse the induced dyslipidemia [124]. Higher HDL levels with tofacitinib are associated with decreased adverse cardiovascular events in patients with RA via a proposed mechanism of alterations to HDL cholesterol fractions that convert them to become anti-inflammatory [125]. There is an inverse correlation between inflammation (as measured by CRP levels) and the levels of small-sized atheroprotective HDL particles [126]. Treatment of RA patients with JAK inhibitors was found to increase the levels of small-sized, atheroprotective HDL molecules. Interestingly, an open-label, prospective study of 46 patients with active RA treated with tofacitinib for 54 weeks resulted in decreased measures of atherosclerosis, including carotid intima-media thickness and pulse-wave velocity [127]. An in vitro study using THP-1 human macrophages showed that tofacitinib decreases intracellular cholesterol and the formation of foam cells by promoting cholesterol efflux to extracellular acceptors such as apoA-I, HDL, and whole normolipidemic human serum [128]. This effect may come about through several mechanisms, one of which is an increase in ABCA1. In addition, based on prior work showing that JAK-2 inhibition in human macrophages downregulates acyl-CoA cholesterol acyltransferase (ACAT), the authors postulate that another mechanism by which tofacitinib may improve cholesterol efflux is by enhancing ACAT expression and thus promoting cholesterol efflux [129]. As noted in this study, tofacitinib decreased intracellular cholesterol in the serum of patients with active inflammatory disease (specifically juvenile idiopathic arthritis), despite the inflammatory environment and defective lipoproteins. A small prospective observational study of microvascular and macrovascular function in 11 RA patients at baseline and 3 months after initiation of JAK inhibitor therapy showed no change in macrovascular structure or function but a significant change in capillaroscopic parameters consistent with vascular impairment [130]. The authors urge further investigation to confirm their findings and in order to assess the ramifications for the long-term cardiovascular impact. Another aspect to be considered in overall RA treatment is the steroid-sparing effect of using JAK inhibitors since reducing the use of glucocorticoids has cardiovascular benefits [131].

### 4.8. Abatacept

Abatacept is a selective co-stimulation modulator. Its mechanism of action is blocking T-cell activity via the CD80/CD86-CD28 costimulatory signal [132]. Treatment with abatacept has been associated with increased insulin sensitivity, which could mitigate the risk of CVD in RA patients with diabetes [133]. Compared to TNF inhibitors, Jin et al. found abatacept to be more protective against a new cardiovascular event and was not associated with higher venous thromboembolic events [134]. In an open-label, prospective, observational study from Japan, Yamada et al. also saw favorable effects of abatacept on the intima-media thickness of the common carotid artery in RA patients treated with abatacept, with smaller increases in the plaque score with abatacept [135]. However, other studies have shown no advantage when comparing biologic treatments and the risk of acute coronary syndrome [136]. Abatacept’s use appears to be generally safe and unlikely to increase the incidence of cardiovascular events [114,135]. Recently, synovial power Doppler ultrasound signals (PDUSs) have been studied in relation to abatacept treatment. Patients treated with abatacept for 3 months had greater decreases in PDUSs and improvement in HDL function compared to those with a lesser change in PDUSs [137]. For patients who were naïve to treatment with biologics, a PDUS signal was associated with significant antioxidant activity of HDL. Conversely, patients with the worst HDL function had greater synovitis. Furthermore, in a population of seropositive patients with moderate to severe RA, abatacept (as compared to adalimumab) had both better clinical efficacy and lower cost per responder due to its higher clinical efficacy [138].

### 4.9. TNF-α Inhibitors

Tumor necrosis factor alpha (TNF-α) inhibitors are a class of biologic medications that work by blocking TNF-α, a protein that causes inflammation and can lead to autoimmune diseases [139]. TNF-α also plays a pivotal role in inflammation and the development of atherosclerosis. Despite efficacy in treating joint disease, the effect of TNF inhibitors on cardiovascular risk remains a topic of debate [140]. Data from a Swedish registry demonstrated a mixed effect of TNF inhibitors on cardiovascular outcomes. Following the initiation of RA therapy, the efficacy of treatment can be assessed in patients via the European League Against Rheumatism (EULAR) response, in which a good response is defined as a significant change in the disease activity score (DAS)28 from baseline. RA patients with a good response to TNF inhibition at 5 ± 3 months exhibited a 40–50% decrease in risk for acute coronary syndrome at 1 year. RA patients who were non-responders and did not have a significant change in DAS28 had rates of acute coronary syndrome at 1 year, more than two times higher than the age- and sex-matched general population [141]. Hussain et al. hypothesized that the varying effects of TNF inhibition on the cardiovascular system are due to the differing roles of tumor necrosis factor receptor 1 (TNFR1) and tumor necrosis factor receptor 2 (TNFR2) in cardiac tissue [142]. TNFR1 is an apoptotic receptor, and its inhibition by TNF-α inhibitors is subsequently cardioprotective. However, TNFR2 is primarily cardioprotective, and its greater inhibition may explain the cardiovascular morbidity associated with TNF-α inhibitors. A systematic review undertaken by Nair et al. found that treatment of RA patients with a TNF-α inhibitor did reduce cardiovascular events, including myocardial infarction, stroke, transient ischemic attack, and coronary artery disease [143]. Other studies have not shown a lower incidence of MI with TNF inhibitor use compared with RA patients treated with traditional DMARDs [144]. This finding may be explained by the neutral effect of TNF inhibition on cholesterol transport at the level of the macrophage. In work from our lab using cultured human macrophages, adalimumab did not affect the levels of multiple reverse cholesterol transport protein mRNAs, including ABCA1, ABCG1, and 27-hydroxylase (unpublished data). This was similarly reflected in a lack of change in macrophage lipid efflux with no significant difference in effluxed cholesterol as a percent of the total (intracellular and extracellular) cholesterol either with or without exposure to adalimumab (U = 19, *p* = 0.5350, *n* = 6 per group). TNF inhibitors may counteract atherosclerosis development in patients with RA by reducing levels of cell surface heparan sulfate proteoglycans and improving endothelial dysfunction [145,146,147].

### 4.10. Tocilizumab

Tocilizumab is a humanized anti-IL-6 receptor antibody that binds to both soluble and membrane-bound IL-6 receptors. Its mechanism of action is the inhibition of IL-6-mediated signaling through these receptors [148]. IL-6 is associated with plaque destabilization, microvascular flow dysfunction, and adverse outcomes in the setting of acute ischemia [149]. Theoretically, IL-6 inhibition should decrease cardiovascular risk. The findings in the literature are mixed. One systematic review and meta-analysis of 14 cohort studies found that tocilizumab was associated with a lower risk of MACE as compared to TNF inhibitors [150]. The ENTRACTE trial, a randomized controlled trial investigating the cardiovascular safety of tocilizumab versus etanercept, found no significant difference in the number of MACE events between the two groups [151]. The tocilizumab group had a median 11.1% greater increase in LDL cholesterol levels, a median 5.4% greater increase in HDL cholesterol levels, and a median 13.6% greater increase in triglyceride levels compared to the etanercept group at 4 weeks (each *p* < 0.001), which persisted over time. Tocilizumab-induced hyperlipidemia has been shown in multiple other studies, without a corresponding increase in cardiovascular risk [152,153].

### 4.11. Rituximab

Rituximab is a chimeric anti-CD-20 monoclonal antibody that exerts its mechanism of action via the depletion of B cells [154]. It promotes both antibody-dependent and complement-dependent cytotoxicity along with upregulating B cell apoptosis, though the exact mechanism underlying its effectiveness in RA treatment is not wholly understood [155]. It is an approved therapeutic agent in RA for patients who do not respond to DMARD therapy. Its efficacy has been established by various studies. Rituximab has been found to significantly improve patient-reported symptoms, decrease joint damage confirmed by radiographic imaging, and initiate remission for some patients, in one case for almost a decade [154]. The side-effect profile of rituximab mainly concerns developing or reactivating infections as well as the decreased function of immunizations. Rituximab has not been shown to increase the risk of MACE events or cause cardiac dysfunction [155]. An open-label longitudinal study found that there were significant reductions in inflammatory markers, disease activity scores, and pulse-wave velocities over time in patients who received rituximab [156]. Improvement in endothelial function can be seen after a single dose of rituximab within 24 weeks [157]. Patients who responded clinically to rituximab were also noted to have improved lipid profiles, decreased arterial stiffness, and carotid intima-media thickness, whereas non-responders did not have significant changes in the aforementioned values [158]. Whether these changes culminate in a clinically significant decrease in cardiovascular risk remains to be seen.

### 4.12. Statins

Statins inhibit the enzyme HMG-CoA reductase and block the conversion of HMG-CoA to mevalonic acid. This reduces hepatic cholesterol synthesis and increases cell surface LDL receptor expression [159]. In the general population, statins are used to decrease the risk of cardiovascular events. The role of statins in mitigating the adverse effects of dyslipidemia in RA is not clear. Statins decrease vascular inflammation with postulated mechanisms involving downregulated IL-6 expression and lowering of homocysteine levels [160,161,162,163]. However, in the RA patient, this is complicated by the lipid paradox. The lipid paradox refers to the association between low lipid levels and active inflammation, leading to worse symptoms. For instance, patients with high disease activity and elevated inflammatory markers often have low LDL and total cholesterol, though with a higher risk of CVD [164]. Studies such as the Trial of Atorvastatin for the Primary Prevention of Cardiovascular Events in Patients with Rheumatoid Arthritis (TRACE RA), which was the largest academically led RA clinical trial, have not been able to establish whether statin therapy would benefit RA patients for primary prevention [165]. The findings from TRACE RA suggest that RA patients experience the same benefits for cardiovascular protection as the general population. However, RA patients had lower rates of cardiovascular events than expected, and this caused the study to be terminated early. Therefore, this study did not support using statins as primary prevention against cardiovascular events in RA patients. The decision to prescribe statins should be made on a case-by-case basis, taking into account the full clinical picture. Smaller studies have shown that statins reduce arterial stiffness and carotid plaque; moreover, other studies have found that stopping statins was associated with poorer outcomes [166]. High-intensity statins not only stopped the formation of new atherosclerotic plaques but they have also been shown to promote the regression of existing plaques as well as encourage the calcification and stabilization of plaques, thereby decreasing the long-term cardiovascular risk of RA patients [167]. Further trials with large sample sizes and longer follow-ups are needed to better define the role of statins in the management of CVD in RA.

**Table 1 biomedicines-12-01608-t001:** Medications used in rheumatoid arthritis and their effect on cardiovascular risk.

Medication Name or Category	Net Effect on Cardiovascular Risk	Putative Mechanism	References
Methotrexate	Beneficial	Increased adenosine via inhibition of adenosine deamination	[74,75,76,77,78,79,80,81]
NSAIDs	Harmful	Inhibition of cyclooxygenase enzymes, interferes with reverse cholesterol transport	[83,84,85,86,87,88]
Leflunomide	Neutral	Inhibition of dihydroorotate dehydrogenase	[89,90,91,92]
Sulfasalazine	Likely Beneficial	Decreased platelet aggregation, inhibition of NF-κB, improved endothelial function	[74,94,95,96,97,98,99,100]
Hydroxychloroquine	Beneficial	Overall reduction in inflammation, possibly anti-thrombotic	[101,102,103,104,105,106,107,108,109]
Glucocorticoids	Harmful	Increased oxidative stress and decreased nitric oxide. Increased adiposity and insulin resistance.	[85,110,111,112,113,114,115,116]
JAK inhibitors	Potential for harm in certain patients	Suppress the JAK-STAT signaling pathway	[117,118,119,120,121,122,123,124,125,126,127,128,129,130]
Abatacept	Neutral	CD80/CD86-CD28 costimulatory modulator	[132,133,134,135,136,137,138]
TNF-α inhibitors	Likely beneficial	Inhibition of TNFR1, amelioration of endothelial dysfunction	[139,140,141,142,143,144,145,146,147]
Tocilizumab	Neutral	Anti-IL-6 receptor antibody, may increase cholesterol levels	[148,149,150,151,152,153]
Rituximab	Neutral	Anti-CD20	[154,155,156,157,158]
Statins	Beneficial	Lower LDL cholesterol, improve lipid profile, anti-inflammatory	[159,160,161,162,163,164,165,166]

Abbreviations: IL—interleukin; JAK-STAT—Janus kinase-signal transducer and activator of transcription; LDL—low-density lipoprotein; NF-κB—nuclear factor-κB; NSAIDs—non-steroidal anti-inflammatory drugs; TNF—tumor necrosis factor; TNFR1—tumor necrosis factor receptor 1.

## 5. Clinical Context

In the absence of established treatment guidelines for CVD in RA, we propose the following screening and treatment paradigm based on the available evidence. One recommendation made by the American Heart Association (AHA) is to assess the risk for atherosclerotic CVD during times of low disease activity because low cholesterol levels during flares of disease can underestimate the atherosclerotic CVD risk [168]. Furthermore, in patients for whom CAD is suspected, coronary computed tomography angiography (CTA), comprised of a non-enhanced and a contrast-enhanced CT scan, may be used to evaluate the coronary artery calcification score to guide treatment. The non-contrast portion of the procedure provides a coronary artery calcium score and, in a large-scale cohort study from Tinggaard et al., a trend toward increased prevalence of calcification was found for seropositive RA patients and patients requiring treatment for relapse or flare [169]. Further compounding the issue of risk assessment in the context of RA is the fact that the atherosclerotic CVD risk calculator is not validated for RA patients and does not consider several factors, for example, the use of glucocorticoids.

Currently, there are no established total cholesterol or LDL goals for RA patients; however, applying goals for diabetic patients may be helpful, considering a similarly elevated risk. The AHA/ACC recommends initiating statin therapy for adults 40 to 75 without diabetes and an intermediate ten-year risk (7.5% to 19.9%) and “risk-enhancing factors”, which include chronic inflammatory disorders such as RA [168]. Inflammation can cause instability and rupture of atherosclerotic plaques, so the risk of myocardial infarction in a patient with RA has been estimated to be similar to patients without RA who have diabetes or are ten years older [170]. Patients with untreated RA or those with a strong disease burden tend to have lower total cholesterol, HDL-C, LDL-C, and triglycerides [171]. This is potentially due to a high level of inflammation limiting the efflux of HDL-C [172]. Treatment to decrease inflammation, in fact, can increase lipid levels but also decrease cardiovascular risk. Total cholesterol had a weak association with myocardial infarction in RA patients as compared to the general population, as found by the AMORIS study, implying that decreasing lipid levels of patients with RA may not be effective as a protective factor against myocardial infarction [173]. However, TRACE RA found that after the initiation of atorvastatin 40 mg daily, there was a decrease of 42% in cardiovascular event risk reduction per mmole/Liter decrease in LDL, yet this treatment did not achieve a significant reduction in cardiovascular risk (hazard ratio [HR] 0.66, 95% CI 0.39–1.11) [165].

Previous sections of this paper have highlighted how inflammatory cytokines such as TNF-α, IL-6, and IL-1β contribute to both the RA disease process and atherosclerosis [46,174] (Figure 2). Observational data suggest that controlling inflammation in RA may favorably affect some CVD risk factors and reduce the development and progression of CVD [175]. The data support the first-line use of methotrexate for RA and its cardiovascular manifestations: among eight large cohorts, methotrexate use was associated with a pooled 28% reduction in CVD event risk [85]. TNF inhibitors are likely a reasonable second choice if methotrexate therapy is contraindicated or insufficient, though evidence for the cardiovascular benefit is mixed. JAK inhibitors, particularly tofacitinib, should be avoided in patients over the age of 65 with other cardiovascular risk factors based on the results of the ORAL surveillance study [118]. Likewise, glucocorticoid and NSAID use in patients with pre-existing cardiovascular risk factors should be avoided given the higher associated risk for adverse cardiovascular events [113,176].

It is also vital to acknowledge how the biases of the studies referred to in this paper may impact the selection of treatment regimens for RA patients in clinical practice. Some limitations for generalizability include the exclusion of diabetic patients in TRACE RA, which may be an important confounding factor as diabetes is a major risk factor for CVD [165]. The predominantly older male population of veterans studied for methotrexate’s impact on CVD risk affects the generalizability to the generally younger female population that is most affected by RA [82]. The ORAL surveillance study could not fully assess the effect of pack years of smoking for RA patients and MACE risk when smoking is a major risk factor for the severity of RA as well as for CVD [118]. This review lays the foundation for clinicians to guide the selection of RA medications based on the existing data (see Table 1).

## 6. Conclusions and Future Perspectives

The mortality gap between RA and non-RA populations due to premature ASCVD represents an urgent research need in the fields of cardiology and rheumatology [177]. A recent study conducted in Norway has shown that patients with RA who are on stable anti-rheumatic therapy that included methotrexate in over 97% of subjects had no elevation in CVD risk above the general population, while the non-stable group had elevated CVD risk [178]. This kind of study may help to distinguish the higher-CVD-risk RA patients who need more intense monitoring [179]. Important areas of focus include a cardiovascular risk assessment tool validated for use in patients with RA. Such a tool could help identify at-risk patients earlier than the widely used Framingham risk assessment score [180]. Artificial intelligence may increase accuracy in ASCVD risk prediction in RA [181]. The lipid paradox in RA, discussed previously, is a second possible area for intervention. A method other than statin therapy is essential to alter pro-inflammatory HDL and inhibit atherosclerosis [67]. Current agents under investigation include those directed at raising the anti-atherogenic lipid apo-A1, others that inhibit cholesteryl ester transfer protein (CETP), which increases HDL cholesterol, and bempedoic acid, which inhibits the de novo synthesis of cholesterol and triglycerides [182,183,184]. These approaches have yielded mixed results [185]. Another potential immunotherapeutic target is IL-10, which, along with TGF-β, inhibits the development of atheromas [57,186]. The management of hypertension is also crucial [187]. Furthermore, the development of specialized centers focused on CVD screening and treatment within rheumatic diseases is warranted.

RA is an independent risk factor for coronary artery disease [188,189]. The chronic activation of inflammatory cytokine pathways may be the shared mechanism that leads to both joint symptoms and an increased atherosclerotic burden. Several medications for RA have cardiovascular benefits. However, CVD remains the leading cause of death in patients with RA, despite major advancements in the treatment of joint disease. As we await new developments in preventing and treating CVD in RA patients, rheumatologists, cardiologists, and primary care providers must recognize and convey the heightened risk to patients, aggressively screen for and treat traditional CVD risk factors, and treat RA disease activity to a target of low disease activity or remission.

## Figures and Tables

**Figure 1 biomedicines-12-01608-f001:**
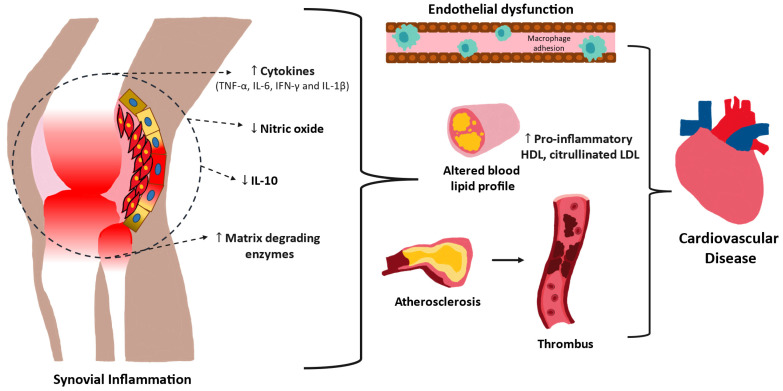
Schematic representation of common mechanisms in the pathogenesis of cardiovascular disease and rheumatoid arthritis. Increased levels of pro-inflammatory cytokines are found in the joint space and the systemic circulation. The combination of pro-inflammatory mediators and reduced levels of atheroprotective nitric oxide and IL-10 trigger endothelial dysfunction. Dysfunctional endothelium, poorly functioning pro-inflammatory HDL, highly atherogenic citrullinated LDL, and an altered blood lipid profile contribute to atherosclerosis and thrombosis, which favor accelerated development of cardiovascular disease in patients with rheumatoid arthritis. ↑ = increases; ↓ = decreases; → = progresses to.

**Figure 2 biomedicines-12-01608-f002:**
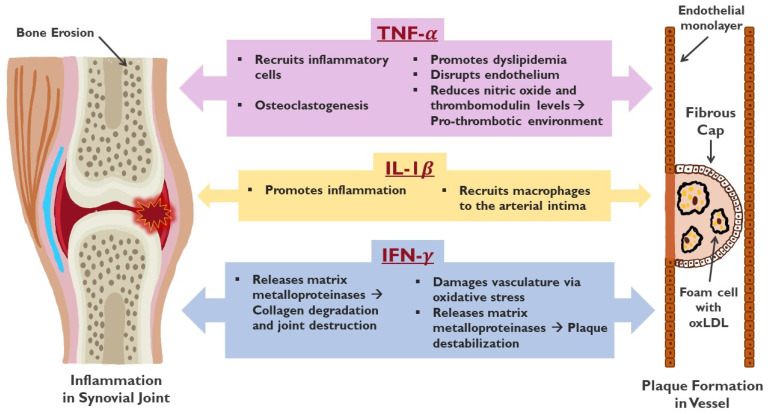
Illustration of the effects of three cytokines, TNF-α, IL-1β, and IFN-γ on the musculoskeletal system (left side) and cardiovascular system (right side).

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
