# Peer review of "Treating Cardiovascular Disease in the Inflammatory Setting of Rheumatoid Arthritis: An Ongoing Challenge"

_biomedicines, 2024, doi:10.3390/biomedicines12071608_

Round 1

Reviewer 1 Report

Comments and Suggestions for Authors

The investigation conducted by the authors authors " Treating Cardiovascular Disease in the Inflammatory Setting of Rheumatoid Arthritis: An Ongoing Challenge " revealed interesting findings. While the study provided valuable insights, there are specific areas requiring further attention, particularly concerning the research focus and outcomes. Nevertheless, substantial revisions are necessary for the manuscript, with several questions and points warranting clarification.

1. the article provides an exploration of the cardiovascular effects of various rheumatoid arthritis (RA) treatments. It discusses multiple treatment options and their potential impacts on heart health.

2. Some sections of the review rely on short-term or observational data, which may not fully capture the long-term cardiovascular effects or risks associated with RA treatments.

3. While the review outlines the mechanisms through which RA treatments may affect cardiovascular health, it may lack depth in discussing the complexities of these pathways or the potential for interactions with other physiological processes.

4. Delve deeper into the mechanisms underlying the cardiovascular effects of RA treatments, discussing potential interactions with other physiological processes and exploring the complexity of these pathways.

5. The review may not comprehensively evaluate the overall risk-benefit profile of each RA treatment concerning cardiovascular health, including considerations such as treatment efficacy, safety, patient preferences, and healthcare costs.

6. How can the review address potential biases and ensure an objective interpretation of the data to provide clear guidance for clinicians regarding the selection and management of RA therapies for patients with heart issues?

Author Response

RE:   Manuscript ID: biomedicines-3012765 Review Article “Treating Cardiovascular Disease in the Inflammatory Setting of Rheumatoid Arthritis: An Ongoing Challenge”

We thank the reviewer for thoroughly scrutinizing our manuscript. As requested, we have revised the manuscript and addressed the specific comments of the reviewer. The revised sections are delineated in red in a marked copy of the manuscript text.

Below, we provide a point-by-point response to the reviewer’s comments.

Reviewer # 1 Comments

  • COMMENT #1: The article provides an exploration of the cardiovascular effects of various rheumatoid arthritis (RA) treatments. It discusses multiple treatment options and their potential impacts on heart health.

RESPONSE: This is the purpose of our review.

  • COMMENT #2: Some sections of the review rely on short-term or observational data, which may not fully capture the long-term cardiovascular effects or risks associated with RA treatments..

RESPONSE: This is true and is inherent in the topic where studies establishing definitive treatment are sadly lacking.

  • COMMENT #3: While the review outlines the mechanisms through which RA treatments may affect cardiovascular health, it may lack depth in discussing the complexities of these pathways or the potential for interactions with other physiological processes.

RESPONSE: We have added depth to this mechanistic overview throughout the manuscript. We have also included a new figure (Figure 1 “Schematic representation of common mechanisms in the pathogenesis of cardiovascular disease and rheumatoid arthritis.”)

  • COMMENT #4: Delve deeper into the mechanisms underlying the cardiovascular effects of RA treatments, discussing potential interactions with other physiological processes and exploring the complexity of these pathways.

RESPONSE: We have added discussion of potential interactions throughout the manuscript.

  • COMMENT #5: The review may not comprehensively evaluate the overall risk-benefit profile of each RA treatment concerning cardiovascular health, including considerations such as treatment efficacy, safety, patient preferences, and healthcare costs.

RESPONSE: We have added discussion of safety, risk and cost within the description of each medication.

  • COMMENT #6: How can the review address potential biases and ensure an objective interpretation of the data to provide clear guidance for clinicians regarding the selection and management of RA therapies for patients with heart issues?

           RESPONSE: We have added a discussion of bias to the Clinical Context section.

We thank the reviewer and believe that the manuscript is improved as a result of their input.  We hope you will agree, and decide in favor of accepting our report at this time.

Reviewer 2 Report

Comments and Suggestions for Authors

The article is very interesting before, it could be considered for further publication I have some queries that authors need to incorporate and revise their Manuscript.

General comments Sentence formation needs crosscheck. Grammatical mistakes need to be minimized.

Abstract section does not give proper information. Abstract means a full-fledged summary that should give readers highlights of the information and topics covered in the manuscript. Please revise the abstract (needs to rephrase and rewrite some sentences). Also, highlight essentialities and future perspectives of the study.

Section Introduction

The authors are advised to develop a proper link between Rheumatoid arthritis with the occurrence of CVDs. Confine it to contents that are concerned with the topic of the manuscript.

Pathophysiology of atherosclerosis

Authors have at once started with the topic of atherosclerosis. They should first make a background to CVDs and then come to the factors, particularly atherosclerosis.

Mechanism and role of inflammation…….

Inflammation in the occurrence of CVDs seems more appropriate for the heading. It would be better to add some figure to the contents of this heading.

Current treatment options…….

Authors should undertake an evaluation of the mechanism of action of each drug. It will add new insights to the story.

Table or figure (as appropriate) should be added.

Clinical context

The contents of the section can be further elaborated.

Future direction and conclusion sections

The sections can be presented together as Conclusion and future perspectives. Authors should make discussion a bit more elaborative and should highlights importance of the study and future directions with possible limitations.

Comments on the Quality of English Language

Manuscript needs to be correction for minor mistakes.

Author Response

RE:   Manuscript ID: biomedicines-3012765 Review Article “Treating Cardiovascular Disease in the Inflammatory Setting of Rheumatoid Arthritis: An Ongoing Challenge”

We thank the reviewer for thoroughly scrutinizing our manuscript. As requested, we have revised the manuscript and addressed the specific comments of the reviewer. The revised sections are delineated in red in a marked copy of the manuscript text.

Below, we provide a point-by-point response to the reviewer’s comments.

Reviewer # 2 Comments

  • COMMENT #1: General comments Sentence formation needs crosscheck. Grammatical mistakes need to be minimized.?

RESPONSE: We have performed a crosscheck and corrected the grammar.

  • COMMENT #2: Abstract section does not give proper information. Abstract means a full-fledged summary that should give readers highlights of the information and topics covered in the manuscript. Please revise the abstract (needs to rephrase and rewrite some sentences). Also, highlight essentialities and future perspectives of the study.

RESPONSE: We have rewritten the abstract as suggested.

  • COMMENT #3: Section Introduction: The authors are advised to develop a proper link between Rheumatoid arthritis with the occurrence of CVDs. Confine it to contents that are concerned with the topic of the manuscript.

RESPONSE: We have rewritten the introduction.

  • COMMENT #4: Pathophysiology of atherosclerosis: Authors have at once started with the topic of atherosclerosis. They should first make a background to CVDs and then come to the factors, particularly atherosclerosis.

RESPONSE: We have now provided a background on CVD.             

  • COMMENT #5 Mechanism and role of inflammation…….Inflammation in the occurrence of CVDs seems more appropriate for the heading. It would be better to add some figure to the contents of this heading.

RESPONSE: We have reworded as suggested and added a figure.

  • COMMENT #6: Current treatment options…….Authors should undertake an evaluation of the mechanism of action of each drug. It will add new insights to the story.

     RESPONSE: We have added this to the content.

  • COMMENT #7: Current treatment options…….Authors should undertake an evaluation of the mechanism of action of each drug. It will add new insights to the story. Table or figure (as appropriate) should be added.

     RESPONSE: We have added this to the content.

  • COMMENT #8: Clinical context: The contents of the section can be further elaborated.

     RESPONSE: We have added more information in this section.

  • COMMENT #9: Future direction and conclusion sections: The sections can be presented together as Conclusion and future perspectives. Authors should make discussion a bit more elaborative and should highlights importance of the study and future directions with possible limitations.

RESPONSE: We have combined and rewritten as advised.

We thank the reviewer and believe that the manuscript is improved as a result of their input.  We hope you will agree, and decide in favor of accepting our report at this time.

Reviewer 3 Report

Comments and Suggestions for Authors

Patients with rheumatoid arthritis (RA) suffer from a significantly increased cardiovascular risk. In this study, Saloni et al. summarize the shared inflammatory pathways between atherosclerosis and rheumatoid arthritis and highlight recent data on the cardiovascular effects of commonly used rheumatoid arthritis medications. While the manuscript aims to present a comprehensive review on cardiovascular disease and rheumatoid arthritis, I found that a portion of the content has been covered in other publications, which diminishes the novelty and depth of this review. Specifically, Chapter 4 - Current Treatment Options, which constitutes the main part of the review, lacks novelty. The organization in this section are similar to those in previously published manuscripts [PMID: 27242206, PMID: 38489782, PMID: 32080804]. Therefore, I do not recommend this manuscript for publication in Biomedicine.

Author Response

RE:   Manuscript ID: biomedicines-3012765 Review Article “Treating Cardiovascular Disease in the Inflammatory Setting of Rheumatoid Arthritis: An Ongoing Challenge”

We thank the reviewer for thoroughly scrutinizing our manuscript. As requested, we have revised the manuscript and addressed the specific comments of the reviewer. The revised sections are delineated in red in a marked copy of the manuscript text.

Below, we provide a point-by-point response to the reviewer’s comments.

Reviewer # 3 Comments

  • COMMENT #1: Patients with rheumatoid arthritis (RA) suffer from a significantly increased cardiovascular risk. In this study, Saloni et al. summarize the shared inflammatory pathways between atherosclerosis and rheumatoid arthritis and highlight recent data on the cardiovascular effects of commonly used rheumatoid arthritis medications. While the manuscript aims to present a comprehensive review on cardiovascular disease and rheumatoid arthritis, I found that a portion of the content has been covered in other publications, which diminishes the novelty and depth of this review. Specifically, Chapter 4 - Current Treatment Options, which constitutes the main part of the review, lacks novelty. The organization in this section are similar to those in previously published manuscripts [PMID: 27242206, PMID: 38489782, PMID: 32080804]. Therefore, I do not recommend this manuscript for publication in Biomedicine.

RESPONSE: We have revised the manuscript extensively and believe it is improved greatly.

PMID: 27242206 was published in 2016 and PMID: 32080804 was published in 2020 and they do not not cover a lot of the newest information. PMID: 38489782 is surgery-focused and looks broadly at an array of rheumatic diseases

We thank the reviewer and believe that the manuscript is improved as a result of their input.  We hope you will agree, and decide in favor of accepting our report at this time.
